# `ParroT`: Translating during Chat using Large Language Models tuned with Human Translation and Feedback

**Wenxiang Jiao**[1*] **Jen-tse Huang**[1,2] **Wenxuan Wang**[1,2] **Zhiwei He**[1,3] **Tian Liang**[1,4]
**Xing Wang**[1] **Shuming Shi**[1] **Zhaopeng Tu**[1]

[1]Tencent AI Lab  [2]The Chinese University of Hong Kong
[3]Shanghai Jiao Tong University  [4]Tsinghua Shenzhen International Graduate School
{joelwxjiao,brightxwang,shumingshi,zptu}@tencent.com
{jthuang,wxwang}@cse.cuhk.edu.hk
zwhe.cs@sjtu.edu.cn  liangt21@mails.tsinghua.edu.cn

## Abstract

Large language models (LLMs) like ChatGPT have exhibited remarkable abilities on a wide range of natural language processing (NLP) tasks, including various machine translation abilities accomplished during chat. However, these models are only accessible through restricted APIs, which creates barriers to new research and advancements in the field. Therefore, we propose ParroT, a framework to enhance and regulate the translation abilities during chat based on open-source LLMs (e.g., LLaMA), human-written translation and feedback data. Specifically, ParroT reformulates translation data into the instruction-following style, and introduces a "Hint" field for incorporating extra requirements to regulate the translation process. Accordingly, we propose three instruction types for finetuning ParroT models, including translation instruction, contrastive instruction, and error-guided instruction. Experiments on Flores subsets and WMT22 test sets suggest that translation instruction improves the translation performance of vanilla LLMs significantly while error-guided instruction can lead to further improvement, which demonstrates the importance of learning from low-quality translations annotated by humans. We also demonstrate the potential of automatic evaluation tools in providing quality information of translations, when constructing error-guided instructions for directions that lack human annotation data. Please refer to our Github project for more implementation details:
https://github.com/wxjiao/ParroT.

## 1 Introduction

Large language models (LLMs), designed in the instruction-following format, such as ChatGPT and GPT-4 (OpenAI, 2023), have garnered considerable interest due to their remarkable abilities in comprehending instructions and generating human-like responses. These versatile models can efficiently perform a wide range of natural language

---
[*]Corresponding author.

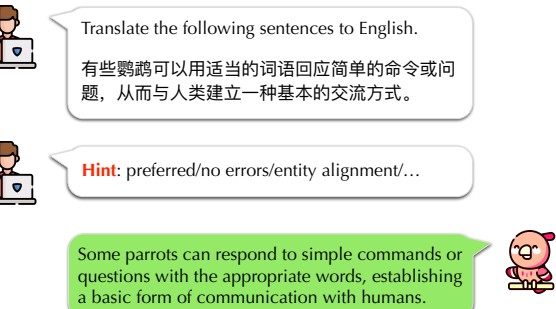

Figure 1: Framework of ParroT. Hints are (optional) extra requirements to regulate the translation process.

processing (NLP) tasks within a single architecture, including question answering (Omar et al., 2023), text summarization (Yang et al., 2023), grammatical error correction (Wu et al., 2023), and machine translation (Jiao et al., 2023). Consequently, they represent a significant stride toward the realization of artificial general intelligence (AGI).

Machine translation, a quintessential NLP task, faces both challenges and opportunities presented by the emergence of LLMs. Traditional machine translation encompasses several sub-tasks (Farhad et al., 2021), such as bilingual translation (Vaswani et al., 2017), multilingual translation (Johnson et al., 2017; Jiao et al., 2022), terminology translation (Wang et al., 2022; Hou et al., 2022), quality estimation (Rei et al., 2020), and automatic post-editing (Pal et al., 2016), among others. These tasks are typically addressed by individual models with limited cross-task interaction. However, current LLMs have the potential to revolutionize this inefficient approach and redefine the machine translation paradigm. On one hand, LLMs can leverage the benefits of various sub-tasks and seamlessly transition between them using only natural language instructions. For instance, if a user is dissatisfied with a translation result, they can request the LLM to refine the translation implicitly (i.e., through automatic post-editing) or explicitly,

by imposing constraints on specific entities (i.e., terminology translation). On the other hand, LLMs are expected to enhance the explainability of machine translation, ultimately leading to further improvements in translation quality. For example, users may want LLMs to compare two translations of a sentence (i.e., quality estimation) and provide an explanation for the discrepancies (i.e., error analysis), which can then be addressed in a targeted manner by the LLM itself. However, superior LLMs like ChatGPT and GPT-4 are only accessible through restricted APIs, which creates barriers to new research and advancements in the field. Therefore, developing comprehensive machine translation abilities upon open-source LLMs has become a critical and challenging research problem.

In this paper, we propose the ParroT framework to enhance and regulate the translation abilities of LLMs during chat by leveraging existing human-written translation and feedback data. To be compatible with chat, our framework reformulates translation data into the instruction-following style (Taori et al., 2023), and introduces a "Hint" field for incorporating extra requirements to guide the translation process. Accordingly, we propose three distinct instruction types: (1) **Translation Instruction**, that asks LLMs to generate translations based on source sentences. (2) **Contrastive Instruction**, that asks LLMs to generate the translations of two different systems with the preferred one at first. (3) **Error-Guided Instruction**, that asks LLMs to generate the translations with human-annotated errors as the hint. The first instruction guarantees the basic translation ability of LLMs while the latter two regulate the LLMs to align with human feedbacks (Ouyang et al., 2022; Liu et al., 2023). We adopt the open-source LLaMA (Touvron et al., 2023) and BLOOM (Scao et al., 2022) models, and conduct instruction tuning on previous WMT validation data and Multidimensional Quality Metric (MQM) human evaluation data. The resulting ParroT models are evaluated on Flores subsets and WMT22 test sets.

Our main findings are summarized as below:

- Translation instruction, as expected, can improve the translation performance of LLMs significantly, especially for directions from English to other languages.

- Error-guided instruction can further improve the performance when asking ParroT to generate translations with no error, indicating the importance of learning from low-quality translations annotated by humans.

- Parameter efficient finetuning with low-rank adaptation (LoRA, Hu et al., 2022) can prevent LLMs from overfitting, which achieves better performance on dominant languages but slows down the learning from other languages.

- We demonstrate the potential of automatic evaluation tools (i.e., COMET) in providing quality information of translations, when constructing error-guided instructions for directions that lack human annotation data.

## 2 Instruction Pool

In this section, we introduce the three distinct instruction types: translation instruction, contrastive instruction, and error-guided instruction. The first instruction guarantees the basic translation ability of LLMs while the latter two regulate the LLMs to align with human-written translation and feedback.

### 2.1 Translation Instruction

As traditional translation systems, we rely on bilingual sentence pairs to accomplish the basic translation ability of LLMs. We follow Stanford Alpaca (Taori et al., 2023) to transform bilingual sentence pairs into the instruction-following format, named **translation instruction**, for finetuning.

Table 1 presents an example of the translation instruction, which includes a preface fixed for all tasks, an "### Instruction:" to describe the translation task (e.g., stating the language pair), an "### Input:" with the source sentence, and a "### Response:" with the target sentence to be generated. To ensure the high quality of sentence pairs, we use human-written translations rather than public training data that could be noisy.

### 2.2 Contrastive Instruction

Besides the basic translation ability, we also want LLMs to understand the relative quality difference between translations. In this way, we may improve the quality of translations by asking LLMs to output the preferred ones. To realize this goal, we need multiple different translations for each source sentence, which can be acquired by the systems submitted to WMT competitions. Meanwhile, the human evaluation results of these systems also provide scores to reflect the quality differences.

Table 1: Instruction types for finetuning ParroT models.

| Type | Instruction |
|---|---|
| | Below is an instruction that describes a task, paired with an input that provides further context. Write a response that appropriately completes the request.\n\n |
| Translation | ### Instruction: \nTranslate the following sentences from Chinese to English.\n\n |
| | ### Input: \n 检查情况显示，市场销售的粮油、肉类、水果、蔬菜、蛋奶等生活必需品供应充足，商品价格基本稳定，未发现严重违法违规行为，市场经营秩序总体平稳。\n\n |
| | ### Response: The inspection results showed that there was an adequate supply of daily necessities, including grain, oil, meat, fruit, vegetable, milk, and eggs in the market and commodity prices basically remain stable, the administration found no serious offensive and noncompliant conducts, and the market order remains stable on the whole. |
| Contrastive | ### Instruction: \nTranslate the following sentences from Chinese to English.\n\n |
| | ### Input: \n 检查情况显示，市场销售的粮油、肉类、水果、蔬菜、蛋奶等生活必需品供应充足，商品价格基本稳定，未发现严重违法违规行为，市场经营秩序总体平稳。\n\n |
| | ### Hint: We prefer to translate it to\n\n |
| | ### Response: <p>The inspection results showed that there was an adequate supply of daily necessities, including grain, oil, meat, fruit, vegetable, milk, and eggs in the market and commodity prices basically remain stable, the administration found no serious offensive and noncompliant conducts, and the market order remains stable on the whole.</p> rather than <p>The results of the inspection indicate the sufficient supply of living necessities on marketing including cereals and oils, meat, fruits, vegetables, eggs and milk, and the basically stabilized commodity price. The inspection hasn't found serious violation of laws and regulations. The market order is stable on an overall basis.</p> |
| Error-Guided | ### Instruction: \nTranslate the following sentences from Chinese to English.\n\n |
| | ### Input: \n 检查情况显示，市场销售的粮油、肉类、水果、蔬菜、蛋奶等生活必需品供应充足，商品价格基本稳定，未发现严重违法违规行为，市场经营秩序总体平稳。\n\n |
| | ### Hint: A translation with major accuracy/mistranslation errors could be\n\n |
| | ### Response: The results of the inspection indicate the sufficient supply of living necessities <v>on marketing</v> including cereals and oils, meat, fruits, vegetables, eggs and milk, and the basically stabilized commodity price. The inspection hasn't found serious violation of laws and regulations. The market order is stable on an overall basis. |
| | ### Hint: A translation with minor fluency/grammar errors could be\n\n |
| | ### Response: The results of the <v>inspection</v> indicate the sufficient supply of living necessities on marketing including cereals and oils, meat, fruits, vegetables, eggs and milk, and the basically stabilized commodity price. The inspection hasn't found serious violation of laws and regulations. The market order is stable on an overall basis. |

As shown in Table 1, we form the response by concatenating two translations (e.g., linked by "rather than"), in which the first translation has a higher quality score. Meanwhile, we indicate that the first translation is preferred in the "### Hint:" field. Essentially, the second translation acts like a negative sample to this sentence pair, which explains the name **contrastive instruction**.

### 2.3 Error-Guided Instruction

The potential problem of contrastive instruction is that, it only tells the LLMs that the two translations have quality differences but not clarify which kind of translation errors lead to such differences. However, we want LLMs to learn the correspondence between the errors and the translations. With such a deeper understanding on the translation errors,

we may ask LLMs to produce translations with no error so as to improve the quality.

We propose **error-guided instruction**. As shown in Table 1, we use the translation with errors annotated by the "<v></v>" span to form the response. Similar to contrastive instruction, we adopt the "### Hint:" field to indicate the error types. This kind of fine-grained error annotation also comes from the human evaluation data.

## 3 Experimental Setups

### 3.1 Training Data

**Alpaca Data.** This dataset is built by Stanford Alpaca (Taori et al., 2023)[1] project, which contains 52.0K instruction-following data of multi-tasks for

---

[1] https://github.com/tatsu-lab/stanford_alpaca

tuning the LLaMA (Touvron et al., 2023)[2] models. We call these data **general instructions**, which help the resulting ParroT models to maintain capabilities on general tasks.

**WMT Validation Data.** We use human-written validation data from previous WMT competitions rather than public training data to avoid introducing noises into instruction tuning. In this version, we use the newstest2017-2020 of Chinese⇔English (i.e., Zh⇔En) and German⇔English (i.e., De⇔En) tasks, which consist of 51.2K sentence pairs for all the four directions. These sentence pairs are formed into the **translation instructions**.

**MQM Human Evaluation Data.** Our human feedback data comes from the Multidimensional Quality Metrics (MQM) datasets (Freitag et al., 2021)[3], which annotate the different translation errors (e.g., major accuracy/mistranslation, minor fluency/grammar) of top WMT systems. Due to its higher reliability than Direct Assessment, MQM was introduced to WMT competitions starting from WMT20 but only provided for a few language pairs. In this version, we use the MQM data for the WMT20 En⇒De and Zh⇒En submissions. These data are formed into the **contrastive instructions** (i.e., 20K) based on the quality scores and the **error-guided instructions** (i.e., 26K) based on the error annotations, respectively.

**Automatically Assessed Data.** Although the Direct Assessment (DA) data of WMT systems provide scores for language directions that lack MQM data (i.e., De⇒En, En⇒Zh), we find the DA score to be very unreliable as they could be quite different for two similar translations. Instead, we opt for automatic evaluation metrics like COMET to score the translations of WMT systems. We also heuristically determine a rough error level for each translation based on the COMET score, namely, Major Error: [0, 85]; Minor Error: (85, 90]; No Error: (90, 100]. This decision comes in part from the observation that top commercial systems achieve COMET scores of nearly 90 on the Flores subsets (Table 3). Finally, we obtain 24K contrastive instructions and 29K error-guided instructions.

**Note**: To obtain a set of diverse instructions, we use the three instructions in Jiao et al. (2023),

including the one in Table 1, as the seeds to ask GPT-4 (OpenAI, 2023) to paraphrase them. In total, we have 33 different instructions that are randomly combined with the training examples.

## 3.2 Model Training

We conduct our experiments with HuggingFace Transformers[4] on open-source LLMs from both the LLaMA family (Touvron et al., 2023) and the BLOOM family (Scao et al., 2022). Specifically, we choose LLaMA-7b and BLOOMZ-7b1-mt with matched parameters, and also include LLaMA-13b and BLOOMZ-560m to study the effect of model sizes. We finetune them to the following variants:

- **Alpaca**, as a reimplementation of the Stanford Alpaca model fine-tuned only on the Alpaca multi-task dataset.

- **ParroT-T**, finetuned on the Alpaca multi-task dataset and only the translation instructions from WMT validation data.

- **ParroT**, finetuned on the Alpaca multi-task dataset, and all the three types of instructions introduced above.

- **ParroT-LoRA**, finetuned by low-rank adaptation (LoRA) with default hyper-parameters from `alpaca-lora`[5], which results in only 4.2M tunable parameters based on LLaMA-7b.

The hyper-parameters for finetuning are basically consistent with Stanford Alpaca (Taori et al., 2023). We finetune the Alpaca and ParroT-T models for 3 epochs on the corresponding data combination. For ParroT and ParroT-LoRA, we finetune them for 1.5 epochs to maintain similar training steps as ParroT-T. We conduct finetuning on 8 Nvidia A100 GPUs and utilize DeepSpeed[6] ZeRO stage 3 for model parallel.

## 3.3 Evaluation

**Test Data.** We evaluate the translation performance of LLMs on two sources of test sets:

- **Flores Subset**: This dataset is a subset of Flores benchmark, in which 50 sentences are sampled for German, English, Romanian and Chinese, respectively, for evaluating the translation performance of ChatGPT (Jiao et al., 2023)

---

[2] https://github.com/facebookresearch/llama
[3] https://github.com/google/
wmt-mqm-human-evaluation

[4] https://github.com/huggingface/transformers
[5] https://github.com/tloen/alpaca-lora
[6] https://github.com/microsoft/DeepSpeed

Table 2: Ablation study of key factors on Flores En⇒De subset with Alpaca based on LLaMA-7b.

| Prompt | Instruct. | Search | BLEU | COMET |
|---|---|---|---|---|
| no-input | TP1 | sample | 20.0 | 80.0 |
| | | beam 4 | 22.1 | 79.1 |
| | TP3 | sample | 19.4 | 79.0 |
| | | beam 4 | 21.5 | 79.0 |
| input | TP1 | sample | 21.0 | 79.5 |
| | | beam 4 | **23.3** | **80.5** |
| | TP3 | sample | 19.3 | 78.6 |
| | | beam 4 | 20.6 | 80.0 |

- **WMT22 Test Sets**: We also use the test sets from WMT22 competition (Kocmi et al., 2022), which are constructed based on more recent content from various domains, including news, social, e-commerce, and conversational domains. The numbers of samples for De⇒En, En⇒De, Zh⇒En and En⇒Zh tasks are 1984, 2037, 1875 and 2037, respectively.

For models based on BLOOM, we only evaluate them on WMT22 test sets since the Flores benchmark has been used in the development of BLOOMZ models.

**Metrics.** For automatic evaluation, we adopt BLEU (Papineni et al., 2002) implemented in SacreBLEU (Post, 2018)[7], and COMET (Rei et al., 2020)[8] from Unbabel/wmt22-comet-da, which are driven by $n$-gram similarity and cross-lingual pretrained models, respectively.

## 4 Results

### 4.1 Ablation Study

Before diving into more experiments, we investigate some factors that may affect the translation performance of LLMs. By default, we conduct the ablation studies on the Flores En⇒De subset with the Alpaca model based on LLaMA-7b.

**Prompt Format.** In the Alpaca multi-task dataset, about 60% examples contain empty "### Input:", which results in two different prompt formats during finetuning, i.e., prompt-input and prompt-no-input. During inference, they use prompt-no-input which combines the instruction and input to fill the "### Instruction:" field, introducing the inconsistency between finetuning and

inference. Therefore, we study if such an operation makes any performance variation.

**Instruction Variation.** Recent studies (Jiao et al., 2023; Zhang et al., 2023) suggest that LLMs are sensitive to task instructions, which could vary the translation performance considerably. We conduct a brief study for this by comparing the TP1 and TP3 instructions in Jiao et al. (2023). TP1 is the one presented in Table 1 while TP3 is "Please provide the [TGT] translation for the following sentences.", which was demonstrated a better choice when tested on ChatGPT[9].

**Search Algorithm.** In machine translation, the beam search strategy (Sutskever et al., 2014; Freitag and Al-Onaizan, 2017; Vaswani et al., 2017) has been the standard search algorithm for inference. However, beam search requires high computation costs which becomes infeasible with the LLMs, since they can easily induce out-of-memory (OOM) issues. Therefore, more efficient search algorithms such as sampling may have to be the choice. Therefore, we compare the sampling strategy (Taori et al., 2023) and the beam search strategy with a beam size of 4 for this factor.

Table 2 presents the results of these ablation studies. We have the following observations: (1) The prompt-input performs slightly better than prompt-no-input though the gap is marginal. (2) The TP1 instruction works better on Alpaca than TP3 which is different from that on ChatGPT. (3) Generally, beam search outperforms sampling significantly, especially in terms of BLEU score. Therefore, we use prompt-input + TP1 + beam search as the default setting for inference.

### 4.2 Main Results

Table 3 and Table 4 present the translation performance of LLaMA and BLOOM models on the test sets. For Flores subsets, we include the baseline results reported in Jiao et al. (2023).

**Instruction tuning exploits the potential of vanilla LLMs for machine translation.** Table 3 shows that the vanilla LLaMA-7b without any further training performs badly on the Flores subsets. By inspecting the outputs, we find that the vanilla LLaMA-7b model tends to generate very long sentences (e.g., copy the instructions, continuing text expansion), which makes the generated text not

---

[7] https://github.com/mjpost/sacrebleu
[8] https://github.com/Unbabel/COMET
[9] https://chat.openai.com

Table 3: Translation performance of LLaMA models on Flores subsets and WMT22 test sets.

| System | De⇒En | | En⇒De | | Zh⇒En | | En⇒Zh | |
|---|---|---|---|---|---|---|---|---|
| | BLEU | COMET | BLEU | COMET | BLEU | COMET | BLEU | COMET |
| **Flores Subsets** | | | | | | | | |
| Google | 45.0 | 88.7 | 41.1 | 88.6 | **31.6** | **87.7** | 43.5 | **88.4** |
| DeepL | **49.2** | **89.7** | 41.4 | 89.0 | 31.2 | 87.3 | **44.3** | 88.1 |
| ChatGPT | 43.7 | 89.1 | 38.8 | 88.1 | 24.7 | 85.8 | 38.2 | 86.9 |
| GPT-4 | 46.0 | 89.3 | **45.7** | **89.2** | 28.5 | 87.4 | 42.5 | 88.4 |
| *Base Model: LLaMA-7b* | | | | | | | | |
| Vanilla | 3.4 | 60.1 | 2.4 | 49.0 | 1.8 | 53.7 | 0.1 | 47.6 |
| Alpaca | 36.6 | 86.8 | 23.3 | 80.5 | 15.1 | 81.2 | 9.8 | 58.6 |
| Alpaca-LoRA | 40.7 | 87.7 | 24.6 | 84.0 | 16.4 | 81.5 | 14.5 | 70.5 |
| ParroT-T | 41.3 | 87.7 | 28.5 | 83.3 | 19.5 | 83.1 | 24.7 | 79.9 |
| ParroT | 41.0 | 87.9 | 30.8 | 84.3 | 19.2 | **83.9** | 25.8 | 80.1 |
| + Infer w/ Prefer. | 38.1 | 87.6 | 23.0 | 83.9 | 18.6 | 83.1 | 22.5 | 80.1 |
| + Infer w/ No Err. | 42.2 | **88.7** | **32.1** | 84.9 | **21.5** | 83.7 | **27.4** | **81.8** |
| ParroT-LoRA | **43.8** | 88.3 | 29.0 | 84.9 | 16.9 | 80.6 | 14.8 | 71.5 |
| + Infer w/ No Err. | 42.0 | 88.0 | 29.8 | **85.4** | 17.4 | 81.3 | 19.8 | 76.7 |
| **WMT22 Test Sets** | | | | | | | | |
| Google | 33.3 | 84.8 | **38.4** | 87.1 | **28.6** | 80.9 | **49.9** | **87.4** |
| DeepL | 32.8 | 84.7 | 36.2 | **87.9** | 24.2 | 79.3 | 44.5 | 86.4 |
| GPT-4 | **33.4** | **84.9** | 34.5 | 87.4 | 24.8 | **82.3** | 41.3 | 87.0 |
| *Base Model: LLaMA-7b* | | | | | | | | |
| Vanilla | 2.9 | 52.8 | 1.6 | 45.3 | 1.2 | 50.3 | 0.3 | 46.3 |
| Alpaca | 27.8 | 82.3 | 20.1 | 78.1 | 14.2 | 74.0 | 10.4 | 62.1 |
| Alpaca-LoRA | 28.9 | **83.2** | 22.1 | 81.3 | 16.1 | 75.6 | 16.3 | 70.6 |
| ParroT-T | 26.6 | 82.5 | 24.0 | 80.4 | 18.1 | 75.3 | 27.0 | 78.4 |
| ParroT | 27.3 | 82.4 | 24.6 | 81.2 | 18.9 | 75.2 | 28.1 | 79.3 |
| + Infer w/ No Err. | 27.3 | 82.4 | **26.1** | 81.6 | **20.2** | **75.9** | 30.3 | 80.3 |
| ParroT-LoRA | 28.8 | 82.8 | 24.0 | 81.4 | 18.2 | 74.7 | 19.9 | 73.7 |
| + Infer w/ No Err. | **29.8** | 83.0 | 24.8 | **81.6** | 19.2 | 75.0 | 20.7 | 74.5 |
| *Base Model: LLaMA-13b* | | | | | | | | |
| Alpaca | 29.7 | 83.1 | 21.4 | 79.4 | 16.2 | 75.9 | 17.6 | 70.8 |
| ParroT | 27.6 | 83.2 | 27.0 | **82.8** | 19.9 | 75.8 | 30.9 | **81.1** |
| + Infer w/ No Err. | **31.1** | 83.6 | **28.1** | 82.6 | **21.7** | **76.7** | 31.7 | 81.0 |

faithful to the source sentences and also not grammatically correct. The reason could be the long context modeling during pretraining. Another reason is that we use the Alpaca inference format, which is basically a zero-shot setting that exhibits no guidance for translation.

Tuning LLaMA-7b on the Alpaca multi-task dataset (i.e., Alpaca) can ameliorate the above issue, resulting in complete generations with proper lengths. We find that Alpaca performs much better on translation, which may benefit from the 0.5% translation instructions in the Alpaca multi-task dataset. However, the best performance is mainly observed on high-resource directions like De⇒En, due to the dominant language of Alpaca dataset in English. Further introducing a small amount of translation instructions (i.e., ParroT-T) in the four language directions can significantly improve the performance, especially for En⇒Zh, in which Chi-

nese was unseen in the pretraining of LLaMA models (Touvron et al., 2023). The findings of these LLaMA-based models are also consistent with that on the WMT22 test sets.

**Learning from low-quality translations annotated by humans is also important.** While presenting the high-quality bilingual pairs to LLMs is important, as discussed above, we argue that low-quality translations annotated by humans also bring benefits. As shown in Table 3, without hint in inference, ParroT outperforms ParroT-T slightly on translation directions from English to other languages (i.e., En⇒De, En⇒Zh). However, when asking ParroT to generate translations **with no error**, the performance can be significantly improved across translation directions and test sets. We speculate that ParroT does learn the relationship between errors and translations by error-guided in-

Table 4: Translation performance of BLOOM models on WMT22 test sets.

| System | De⇒En | | En⇒De | | Zh⇒En | | En⇒Zh | |
|---|---|---|---|---|---|---|---|---|
| | BLEU | COMET | BLEU | COMET | BLEU | COMET | BLEU | COMET |
| *Base Model: BLOOMZ-560m* | | | | | | | | |
| Alpaca | 4.4 | 55.2 | 0.5 | 30.8 | 6.9 | 70.1 | 2.0 | 54.0 |
| ParroT | 16.4 | 68.9 | 13.3 | 57.7 | 16.0 | 74.8 | 25.4 | 79.0 |
| + Infer w/ No Err. | **16.9** | **69.3** | 12.8 | 56.8 | 15.7 | **75.0** | 26.3 | **79.5** |
| *Base Model: BLOOMZ-7b1-mt* | | | | | | | | |
| Alpaca | 17.6 | 73.0 | 3.1 | 44.5 | 13.0 | 76.4 | 23.9 | 81.8 |
| ParroT | 23.1 | 77.6 | 20.0 | 72.7 | 21.4 | 78.5 | 32.4 | **83.6** |
| + Infer w/ No Err. | **24.9** | **78.0** | **20.5** | **73.6** | **22.7** | **79.0** | **34.5** | 83.5 |

Table 5: Effects of error levels as hints during inference. Red : improvement; Green : degradation.

| Hint | En⇒De | | Zh⇒En | |
|---|---|---|---|---|
| | BLEU | COMET | BLEU | COMET |
| None | 30.8 | 84.3 | 19.2 | 83.9 |
| No Err. | 32.1 | 84.9 | 21.5 | 83.7 |
| Minor Err. | 28.8 | 83.6 | 20.6 | 82.1 |
| Major Err. | 28.5 | 82.9 | 19.3 | 80.5 |

struction, such that it can avoid the translation errors as much as possible when the hint of no error is provided.

A bit unexpected is that when asking ParroT to generate preferred translations, the performance drops considerably. As stated in Section 2.3, contrastive instruction only indicates that two translations may have quality differences but not state why, which is difficult for LLMs to identify by themselves. Previous study by Min et al. (2022) also suggests that it is easier for LLMs to learn the instruction formats rather than the input-response patterns, which may explain the phenomenon here.

**Parameter efficient finetuning may prevent LLMs from overfitting.** We also try low-rank adaptation (LoRA, Hu et al., 2022) to finetune partial parameters of LLMs for efficiency. Experimental results in Table 3 show that Alpaca-LoRA outperforms its full model counterpart noticeably. We speculate that LoRA can prevent LLMs from overfitting the small Alpaca multi-task dataset, leading to a stronger generalization ability. However, applying LoRA to ParroT exhibits distinct behaviors for high-resource and low-resource translation directions. Specifically, ParroT-LoRA outperforms the corresponding full model ParroT on De⇒En but performs much worse on the other directions. It seems that the small amount of tunable param-

eters also hinder the learning of instructions from other translation directions. Obviously, the hyperparameters of LoRA should also be properly adjusted to better learn from more instruction data.

**LLMs families and sizes also matter.** For both LLaMA and BLOOM families, larger models can achieve much better translation performance after instruction tuning. Our ParroT framework proves to be effective across all the models. Comparing the two LLMs families, the ParroT model based on BLOOMZ-7b1-mt performs much better on Zh⇒En and En⇒Zh directions than those based on LLaMA-7b, which mainly results from the better modeling of Chinese during the pretraining process of BLOOM.

**Automatic evaluation tools can be effective in constructing error-guided instructions.** In Section 3.1, we construct the automatically assessed data for De⇒En and En⇒Zh that are not provided with the MQM data. As shown in Table 3 and Table 4, we can observe considerable improvements of error-guided instruction on these two translation directions. It demonstrates the potential of automatic evaluation tools (i.e., COMET) in providing the quality information of translations, as an augmentation to translation directions that lack human annotation data.

### 4.3 Analysis

We conduct more analyses to understand the effects of our instruction types. By default, we use the model variants based on LLaMA-7b, and the Flores subsets.

**Effectiveness of Error-Guided Instruction.** To understand how error-guided instruction works, we investigate the behavior of ParroT when asking it to generate translations with varied error levels as

Table 6: Examples from Flores Zh⇒En subset when translated with varied error levels. `Red`: translation errors.

| | COMET | Example |
|---|---|---|
| SRC | n/a | 例如，在卢瓦尔河谷和莱茵河谷参观城堡，或者乘游轮去游览多瑙河沿岸好玩的城市，或者沿伊利运河乘船游玩。 |
| REF | n/a | For example visiting castles in the Loire Valley, the Rhine valley or taking a cruise to interesting cites on the Danube or boating along the Erie Canal. |
| None | 81.2 | For example, visiting the castles in the `Rhine Valley` and the `Lahn Valley`, or taking a cruise along the `Rhine River` to explore the cities `along the river`, or taking a boat trip along the `Illi River`. |
| No Err. | 81.4 | For example, visiting the castles in the `Rhine Valley` and the `Ruhr Valley`, or taking a cruise to visit the cities along the `Rhine River`, or taking a boat ride along the `Illi River`. |
| Minor Err. | 78.1 | For example, visiting the castles in the `Rhine Valley` and the `Ruhr Valley`, or taking a cruise to visit the cities along the `Rhine River`, or taking a boat to `play in the cities` along the `Elbe River`. |
| Major Err. | 65.3 | For example, visiting the `fortress` in the valley of the `River Volga` and the `River Levan`, or taking a cruise to visit the cities along the `River Volga`, or taking a boat to `play in the cities` along the `River Volga`. |

Table 7: Effects of preference as hints during inference. `Red`: improvement; `Green`: degradation.

| Hint | En⇒De | | Zh⇒En | |
|---|---|---|---|---|
| | BLEU | COMET | BLEU | COMET |
| None | 30.8 | 84.3 | 19.2 | 83.9 |
| Prefer. | 23.0 | 83.9 | 18.6 | 83.1 |
| Unprefer. | 29.1 | 83.7 | 19.6 | 82.3 |

hints. As shown in Table 5, the translation quality is getting worse from no error to minor error to major error, especially in terms of COMET score. The translations generated with no hint are usually comparable with the minor error level. It demonstrates that ParroT can place erroneous translations into other locations of the probability space with the regulation of human annotations. As a result, ParroT is more likely to generate high-quality translation with "no error".

For qualitative analysis, we show an example from Flores Zh⇒En subset in Table 6, in which we highlight all errors in each translation. Compared to no error level, minor and major error levels tend to produce more over-translations and mistranslations. It is also important to point out that no error level does not guarantee that completely correct translations will be generated, especially for named entities, which we attribute to the under-explored translation abilities of current LLMs.

**Failure of Contrastive Instruction.** We try to understand why contrastive instruction does not work. By examining the responses of ParroT when asking it to generate preferred translations, we observe significant differences in lexical choices

between the "preferred" and "unpreferred" (i.e., the second translation in the response) translations. Surprisingly, as shown in Table 7, the "unpreferred" translations obtain a much higher BLEU score but the situation is different for the COMET score. It indicates that ParroT attempted to identify the quality differences between the first and second translations in the contrastive instructions through lexical choices, which is a low-level pattern to reflect the translation quality. One potential reason is that the WMT systems are so competitive with each other that the quality differences between them are too subtle for the LLM to learn effectively. We will investigate more about contrastive instruction in future work.

## 5 Related Work

**LLMs for MT.** With the increasing capacity of LLMs, they have become good few-shot learners (Brown et al., 2020; Lin et al., 2022) on various NLP tasks, including machine translation. A number of recent studies focus on how to prompt LLMs for machine translation, including prompt template comparison (Zhang et al., 2023), few-shot example selection (Agrawal et al., 2022; Vilar et al., 2022), domain adaptation (Moslem et al., 2023), and rare word translation (Ghazvininejad et al., 2023). However, our ParroT framework aims to develop instant translation capability for chatbots without few-shot examples. This is consistent with the performance of ChatGPT and GPT-4 (OpenAI, 2023), which exhibit excellent translation ability (Jiao et al., 2023; Bang et al., 2023; He et al., 2023; Liang et al., 2023) during chat.

**Instruction Tuning.** To eliminate the reliance on few-shot examples, recent studies also try to finetune LLMs on a small amount of instructions covering different NLP tasks, making the LLMs zero-shot learners (Mishra et al., 2022; Wei et al., 2022). With the emergence of various powerful open-source LLMs such as BLOOM (Scao et al., 2022) and LLaMA (Touvron et al., 2023), there has been a boom for creating instruction data and tuning customized chatbots, for example, Alpaca (Taori et al., 2023), Vicuna, WizardLM (Xu et al., 2023) and the like. However, most of these studies focus on developing chatbots that are capable of general NLP tasks, while we pay more attention to machine translation. More importantly, apart from the instructions built from parallel translation data, we also transform human feedback data into instructions and demonstrate its effectiveness in improving the translation performance.

## 6 Conclusion

We propose ParroT to enhance and regulate the translation abilities during chat based on open-source LLMs, human-written translation and feedback data. We reformulate translation data into the instruction-following style, and introduce a "Hint" field for incorporating extra requirements to regulate the translation process. Accordingly, we propose three instruction types for finetuning ParroT models, i.e., translation instruction, contrastive instruction, and error-guided instruction. Experiments on Flores subsets and WMT22 test sets suggest that translation instruction improves the translation performance of vanilla LLMs significantly while error-guided instruction can lead to further improvement, demonstrating the importance of learning from low-quality translations annotated by humans. While we only use three instruction types in this paper, it is natural to extend ParroT to other hints (e.g., entity alignments), which we leave for future exploration.

## Limitations

This work performs a preliminary exploration on the instant translation capability for chatbots, which can be further improved in the following aspects:

- **Instruction Variants**: Presently, the instructions only support the translation of incoming sentences. It may be beneficial for chatbots to also translate previous chat records when users struggle to comprehend responses in foreign languages.

- **Contrastive Translations**: In this study, we did not observe performance improvements related to contrastive instructions, possibly due to incorrect instruction formatting. By exploring alternative formats, such as automatic post-editing (APE), we could potentially capitalize on the advantages of contrastive translations.

- **LoRA Effectiveness**: The current analysis did not reveal consistent performance improvements when using LoRA as compared to full model training. It may be necessary to adjust the number of tunable parameters according to the dataset size for better results.

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

## A Flores Full Sets

We adopted the subset of Flores in order to directly compare with the results of ChatGPT and commercial MT systems from the previous evaluation report (Jiao et al., 2023), as stated in Section 3.3. However, for a more convincing evaluation, we also test the LLMs on the full sets of Flores (i.e., with 1012 sentences), and list the results in Table 8. Obviously, the trend of performance across systems still holds, and our "ParroT + Infer w/ No Err" setting performs the best.

## B General Tasks

In our experiments, we include the Alpaca multi-task dataset for finetuning ParroT to preserve its ability on general tasks. We present two examples in Table 9, from question answering and code generation, respectively. The responses of both Alpaca and ParroT are correct, although the expressions may be different. For comprehensive comparison, it is better to perform a quantitative evaluation and we leave it for future work.

Table 8: Translation performance of LLaMA models on Flores full sets.

| System | De⇒En | | En⇒De | | Zh⇒En | | En⇒Zh | |
|---|---|---|---|---|---|---|---|---|
| | BLEU | COMET | BLEU | COMET | BLEU | COMET | BLEU | COMET |
| *Base Model: LLaMA-7b* | | | | | | | | |
| Vanilla | 3.0 | 58.2 | 1.7 | 48.8 | 2.7 | 52.0 | 0.1 | 48.6 |
| Alpaca | 36.3 | 87.6 | 23.0 | 81.1 | 16.4 | 81.3 | 8.3 | 56.5 |
| ParroT-T | 37.4 | 87.9 | 26.9 | 83.7 | 19.7 | 82.8 | 26.1 | 79.2 |
| ParroT | 38.1 | 88.0 | 28.9 | 84.5 | 21.1 | 83.0 | 27.0 | 80.3 |
| + Infer w/ Prefer. | 34.6 | 87.5 | 24.8 | 83.9 | 18.1 | 82.4 | 24.2 | 79.0 |
| + Infer w/ No Err. | **39.4** | **88.4** | **30.7** | **84.6** | **21.3** | **83.4** | **29.1** | **81.2** |

Table 9: Examples of general tasks. The base model is LLaMA-7b.

| System | Question Answering |
|---|---|
| | Tell me about alpacas. |
| Alpaca | Alpacas are a species of camelid native to South America. They are domesticated and bred for their fiber, which is used in the production of knitted and woven items such as clothing, blankets, and rugs. Alpacas are typically kept as livestock, and can live up to 20 years in captivity. They are typically brown, white, or grey in color, and can grow up to 12 inches tall at the shoulder. |
| ParroT | Alpacas are a species of domesticated South American camelids, bred primarily for their fiber. They are typically kept as pets, for their fleece, or for their meat. Alpacas are very social animals and live in herds, with a dominant male leader and a number of females and young. They are very intelligent and can be trained to respond to basic commands. Alpacas require a lot of space and good quality food in order to thrive. |

| System | Code Generation |
|---|---|
| | Write a Python program that prints the first 10 Fibonacci numbers. |
| Alpaca | ```
numbers = [0, 1]
for i in range(2, 10):
        numbers.append(numbers[i-2] + numbers[i-1])

print(numbers)
``` |
| ParroT | ```
def fibonacci(n):
        if n == 0:
                return 0
        elif n == 1:
                return 1
        else:
                return fibonacci(n-1)+fibonacci(n-2)

print(fibonacci(10))
``` |