# OpenReview forum: "ParroT: Translating during Chat using Large Language Models tuned with Human Translation and Feedback"
_EMNLP/2023/Conference — EMNLP 2023 Findings_

### Official Review · Reviewer_AfZp · 2023-08-02

**Soundness:** 4

**Excitement:**

4: Strong: This paper deepens the understanding of some phenomenon or lowers the barriers to an existing research direction.

**Paper Topic And Main Contributions:**

The paper aims at improving machine translation performance of pre-trained LLMs (with the final goal of bringing MT capabilities of open-source LLMs closer to those of models which are only accessible via services, such as ChatGPT); this appears to be a relevant direction of work. Specifically, the paper proposes ParroT which aims at increasing LLM translation capabilities during chat by leveraging human-written translations as well as quality information about translations (specifically human annotations pointing to errors). The information is provided to the LLM via different types of instructions, which include “hints” fields. Specifically, the authors test with using contrastive instructions which comprise 2 translations of different quality, and error-guided instructions which explicitly hint the LLM on errors in example translations. The approach is being evaluated with one LLM mostly trained on English data (i.e. LLaMA) and one multlingual model (i.e. BLOOMZ-7b), indicating that the proposed approach with error-guiding yields improvements over the vanilla approach as well as over using instruction tuning with alpaca dataset. The authors compare full fine-tuning vs fine-tuning with LoRA, indicating interesting difference in performance across language pairs, and some further analyses and interesting insights are provided. However, results are constrained to only very few languages, as data comprising annotations on translation quality/errors appears to be limited, and thus the approach may not be easily applicable to many languages (except if corresponding data are being collected which may be expensive), and it is unclear how well it would work for further languages (given that differences are already being identified for the explored languages).

**Reasons To Accept:**

* The paper is well written and easy to follow
* The approach is interesting, the experiments indicate gains over vanilla baselines, and the paper comprises some interesting analyses and insights.
* Different approaches for fine-tuning are included, i.e. regular FT and LoRA, showing interesting differences in performance w.r.t. language data sources.

**Reasons To Reject:**

* The approach assumes the availability of human labeled data / data which comprises information on error patterns for translations, which appears to be readily available only for few datasets and languages. Scaling to further languages may thus be rather difficult.
* The choice of languages in the experiments is rather constrained (likely due to the previous point), and it is thus unclear how well it would work for further languages (given that differences are already being identified for the explored languages).



**Reproducibility:**

4: Could mostly reproduce the results, but there may be some variation because of sample variance or minor variations in their interpretation of the protocol or method.

**Reviewer Confidence:**

3: Pretty sure, but there's a chance I missed something. Although I have a good feel for this area in general, I did not carefully check the paper's details, e.g., the math, experimental design, or novelty.

---

> ### Author Rebuttal · Authors · 2023-08-29
>
> We appreciate the reviewer's efforts in reviewing our paper.
>
> Concerning the question of scaling languages, we would like to share our ideas with the reviewer.
>
> 1. It is true that human annotation data is readily available only for few languages. However, with the rapid development of LLMs, human annotators have been much more involved, making it feasible to obtain human evaluation data for other languages.
> Besides, we also demonstrate the potential of automatic evaluation tools (e.g., COMET) in providing the quality information of translations, as an augmentation to translation directions that lack human annotation data. In Section 3.1, we constructed the Automatically Assessed Data for De->En and En->Zh, which do not have the MQM data. In Table 3, we can also observe considerable improvements of error-guided instruction on De->En and En->Zh.
> In addition, recent studies on LLMs as evaluators (e.g., AutoMQM [1]) can also serve as potential alternatives to label the error patterns of translations for more languages.
>
>
> 2. In this work, we focus on investigating different strategies of instruction-following on LLMs for MT, and conduct experiments on two representative language pairs (i.e., De-En and Zh-En) that have been widely used in WMT competitions and MT researches. Generally, our error-guided instructions can bring the most improvements over baselines across different translation directions. We believe our approach can generalize to more languages and plan to investigate it in the near future.
>
> [1] The Devil is in the Errors: Leveraging Large Language Models for Fine-grained Machine Translation Evaluation. Fernandes et al. arXiv 2023.

---

### Official Review · Reviewer_jRLd · 2023-08-05

**Soundness:** 4

**Excitement:**

3: Ambivalent: It has merits (e.g., it reports state-of-the-art results, the idea is nice), but there are key weaknesses (e.g., it describes incremental work), and it can significantly benefit from another round of revision. However, I won't object to accepting it if my co-reviewers champion it.

**Paper Topic And Main Contributions:**

The paper proposes to instruction tune LLM for machine translation. They experiment with different ways of instruction tuning including instruction for direct translation, instruction for a preferred translation over a less preferred translation (contrastive instruction), and instruction for controlling the amount of error in the generated translation.
They evaluate with two LLMs: LLaMa 7B, and BLOOMZ-7B parameters. They also compare between full finetuning and LoRA based finetuning.

Contributions:
1. The paper compares between different ways of instruction tuning for machine translation
2. They find that direct instruction tuning and error-guided instruction tuning works well.
3. They find that contrastive instruction is not effective.
4. Also LoRA based finetuning is more effective for high resource language pair

**Reasons To Accept:**

1. The paper does a detailed analysis of what is effective for instruction tuning of LLM for machine translation

**Reasons To Reject:**

1. The paper conducts evaluation for few language pairs including De<->En and Zh<->En
2. In the evaluation on FLORES they show results with a subset of only 50 sentences. Thus the results for FLORES may not be reliable.

**Reproducibility:**

4: Could mostly reproduce the results, but there may be some variation because of sample variance or minor variations in their interpretation of the protocol or method.

**Reviewer Confidence:**

4: Quite sure. I tried to check the important points carefully. It's unlikely, though conceivable, that I missed something that should affect my ratings.

---

> ### Author Rebuttal · Authors · 2023-08-29
>
> We appreciate the reviewer's efforts in reviewing our paper.
>
> 1. Concerning the number of languages, we have two considerations.
>     - While scaling languages is a promising direction for machine translation (MT), it is not our focus in this work. We aims to investigate different strategies of instruction-following on LLMs for MT, and conduct experiments on two representative language pairs (i.e., De-En and Zh-En) that have been widely used in WMT competitions and MT researches.
>
>     - Another reason is the scarcity of existing high-quality human annotation data for MT, as stated in Section 3.1. However, with the rapid development of LLMs, human annotators have been much more involved, making it feasible to obtain human evaluation data for other languages. Besides, we also demonstrate the potential of automatic evaluation tools (e.g., COMET) in providing the quality information of translations, as an augmentation to translation directions that lack human annotation data. In Section 3.1, we constructed the Automatically Assessed Data for De->En and En->Zh, which do not have the MQM data. In Table 3, we can also observe considerable improvements of error-guided instruction on De->En and En->Zh. In addition, recent studies on LLMs as evaluators (e.g., AutoMQM [1]) may also serve as potential alternatives to label the error patterns of translations for more languages.
>
>     Nonetheless, it will be interesting to see how our method performs when extended to more languages. We plan to investigate this in the near future.
>
>
> 2. Thanks for the good advice. We adopted the subset of Flores in order to directly compare with the results of ChatGPT and commercial MT systems from a previous evaluation report, as stated in Section 3.2. However, for a more convincing evaluation, we also test the LLMs on the full set of Flores ( with 1012 sentences), and list the results as below (Base Model: LLaMA-7b). Obviously, the trend of performance across systems still holds, and our "ParroT-Hint + Infer w/ No Err" performs the best.
>
>     | System | De-En (BLEU) | De-En (COMET) |  En-De (BLEU) | En-De (COMET) |  Zh-En (BLEU) | Zh-En (COMET) |  En-Zh (BLEU) | En-Zh (COMET) |
>     | ------ | ------: | ------: | ------: | ------: | ------: | ------: | ------: | ------: |
>     | Vanilla |  3.0 | 58.2 | 1.7 | 48.8 | 2.7 | 52.0 | 0.1 | 48.6 |
>     | Alpaca |  36.3 | 87.6 | 23.0 | 81.1 | 16.4 | 81.3 | 8.3 | 56.5 |
>     | ParroT |  37.4 | 87.9 | 26.9 | 83.7 | 19.7 | 82.8 | 26.1 | 79.2 |
>     | ParroT-Hint |  38.1 | 88.0 | 28.9 | 84.5 | 21.1 | 83.0 | 27.0 | 80.3 |
>     |  + Infer w/ Prefer. |  34.6 | 87.5 | 24.8 | 83.9 | 18.1 | 82.4 | 24.2 | 79.0 |
>     |  + Infer w/ No Err. |  **39.4** | **88.4** | **30.7** | **84.6** | **21.3**  | **83.4** | **29.1** | **81.2** |
>
>
> [1] The Devil is in the Errors: Leveraging Large Language Models for Fine-grained Machine Translation Evaluation. Fernandes et al. arXiv 2023.

---

### Official Review · Reviewer_k66w · 2023-08-11

**Soundness:** 2

**Excitement:**

1: Poor: I cannot identify the contributions of this paper, or I believe the claims are not sufficiently backed up by evidence. I would fight to have it rejected.

**Paper Topic And Main Contributions:**

The authors propose the ParroT, a framework to enhance and regulate the translation abilities during chat based on open-sourced LLMs~(e.g., LLaMA), human-written translation and feedback data.

**Reasons To Accept:**

The topic of this paper is interesting and of broad interest to the community of NLP.

**Reasons To Reject:**

Though the authors proposing a novel framework to incorporate human feedback for improving machine translation, this paper does not sufficiently demonstrate significant advantages over existing methods. The incremental contributions are not convincingly conveyed through thorough quantitative analysis and comparative experiments. More evidence is needed to highlight the novelty beyond incremental improvements to justify publication at top NLP venues.

**Reproducibility:**

3: Could reproduce the results with some difficulty. The settings of parameters are underspecified or subjectively determined; the training/evaluation data are not widely available.

**Reviewer Confidence:**

3: Pretty sure, but there's a chance I missed something. Although I have a good feel for this area in general, I did not carefully check the paper's details, e.g., the math, experimental design, or novelty.

---

> ### Author Rebuttal · Authors · 2023-08-29
>
> We appreciate the reviewer's efforts in reviewing our paper.
>
> We would like to clarify our contributions here.
> 1. We propose a novel framework to incorporate extra requirements into instructions to regulate the translation process. While we mainly introduce instruction types based on human written translations and annotations, the framework is also compatible with other data resources (e.g., entity alignment).
>
> 2. We conduct extensive experiments to compare our instruction strategies (translation instruction, error-guded instruction, and contrastive instruction) with the prevelant strategies (i.e., Alpaca type). We demonstrate the effectiveness of our instructions across four translation directions, upon two LLMs (i.e., LLaMA-7b and BLOOMZ-7b1-mt), with full model finetuning and LoRA tuning.
>
> 3. We also demonstrate the potential of automatic evaluation tools in providing the quality information of translations, as an augmentation to translation directions that lack human annotation data. In Section 3.1, we constructed the Automatically Assessed Data for De->En and En->Zh, which do not have the MQM data. In Table 3, we can also observe considerable improvements of error-guided instruction on De->En and En->Zh.
>
> Nonetheless, there could be other aspects to further improve this paper. We would appreciate if the reviewer can suggest some specific directions, and are glad to refine the paper accordingly.

---

### Meta-Review · Area_Chair_vAbK · 2023-09-23

**Recommendation:** 4

**Metareview:**

In this work, the authors propose PARROT, a instruction-guided technique to improve the translation abilities of open-sourced LLMs. Translation data is rewritten in instruction-style in three different ways: 1) Using standard translation instructions 2) Using contrastive instructions and 3) Using error-guided instructions. The authors find translation instructions to be very helpful and error-guided instructions to further improve performance. However, contrastive instructions do not yield performance improvements.They also find LoRA-based finetuning to be more effective for high resource language pairs.

R2 and R3 have both rated this work highly on soundness (4/4). R1 listed a number of limitations including the need for more baseline comparisons (esp with larger models), the difference between error-guided and chain-of-thought based methods and the main motivation behind the work. While R1's comments came quite late during the rebuttal window, the authors have offered a fairly thorough response with additional experiments.

---

### Decision · Program_Chairs · 2023-10-07

**Decision:**

Accept-Findings

**Comment:**

In this work, the authors propose PARROT, a instruction-guided technique to improve the translation abilities of open-sourced LLMs. Translation data is rewritten in instruction-style in three different ways: 1) Using standard translation instructions 2) Using contrastive instructions and 3) Using error-guided instructions. The authors find translation instructions to be very helpful and error-guided instructions to further improve performance. However, contrastive instructions do not yield performance improvements.They also find LoRA-based finetuning to be more effective for high resource language pairs.

R2 and R3 have both rated this work highly on soundness (4/4). R1 listed a number of limitations including the need for more baseline comparisons (esp with larger models), the difference between error-guided and chain-of-thought based methods and the main motivation behind the work. While R1's comments came quite late during the rebuttal window, the authors have offered a fairly thorough response with additional experiments.